# On Masked Pre-training and the Marginal Likelihood

**Pablo Moreno-Muñoz**
Section for Cognitive Systems
Technical University of Denmark (DTU)
pabmo@dtu.dk

**Pol G. Recasens**[*]
CROMAI, Barcelona Supercomputing Center
Universitat Politècnica de Catalunya (UPC)
pol.garcia@bsc.es

**Søren Hauberg**
Section for Cognitive Systems
Technical University of Denmark (DTU)
sohau@dtu.dk

## Abstract

Masked pre-training removes random input dimensions and learns a model that can predict the missing values. Empirical results indicate that this intuitive form of self-supervised learning yields models that generalize very well to new domains. A theoretical understanding is, however, lacking. This paper shows that masked pre-training with a suitable cumulative scoring function corresponds to maximizing the model's marginal likelihood, which is *de facto* the Bayesian model selection measure of generalization. Beyond shedding light on the success of masked pre-training, this insight also suggests that Bayesian models can be trained with appropriately designed self-supervision. Empirically, we confirm the developed theory and explore the main learning principles of masked pre-training in large language models.

## 1 Introduction

Masked pre-training (MPT) is a family of self-supervised learning methods (Dosovitskiy et al., 2020; Devlin et al., 2018; Caron et al., 2021), that empirically has been demonstrated to result in models that generalize very well to new settings. In essence, masked pre-training removes random features of the data and learns a model to recover these from the remaining input. While empirical results are impressive, a deeper understanding of *why* pre-trained models generalize so well is lacking. Is it due to the use of transformer architectures (Vaswani et al., 2017), the vast over-parametrization (Neyshabur et al., 2019), or something entirely different?

The marginal likelihood or *evidence* is commonly used as the measure of generalization ability in Bayesian models (Tenenbaum and Griffiths, 2001; MacKay, 2003). While computationally expensive, the *blessing* of the marginal likelihood comes from the probabilistic integration of hypotheses. Whenever we are considering a latent variable model in the Bayesian framework, such integration can be thought of as the average over all the possible latent variable mappings, weighted by our prior beliefs. Since masked pre-training drives generalization so well, the lingering question in the Bayesian modeling community is then: *Is masked pre-training somehow related to the maximization of the marginal likelihood?*

**In this paper**, we provide a positive answer. We show that masked pre-training optimizes according to a stochastic gradient of the log-marginal likelihood (LML). Importantly, the log-marginal likelihood is equivalent to the cumulative sum of masked pre-training losses shaped with different sizes for the random mask. Even if its practical use avoids this cumulative sum, we demonstrate that choosing a

---

[*]Work done during an *Erasmus* exchange in DTU (Denmark).

37th Conference on Neural Information Processing Systems (NeurIPS 2023).

*fixed* masking rate, e.g. 15% as in BERT (Devlin et al., 2018), leads to a stochastic *biased* estimation which still maximizes the log-marginal likelihood.

**Our proof** relies on a previous observation from Fong and Holmes (2020), who show that the log-marginal likelihood equals the average of exhaustive *leave-$M$-out* cross-validation (CV) given posterior predictive scores. Intuitively, our formal results can be seen as the *transposed* version of Fong and Holmes's results: where CV removes *random observations* to measure generalization, masked pre-training removes *random features*. While the seminal link between CV and the marginal likelihood was purely a formal result that pointed out the underlying presence of Bayesian principles in a well-known class of learning, our work extends the theory behind the marginal likelihood to comprehend the impressive behavior of the latest generative models.

## 2 Masked pre-training

Masked pre-training (MPT) is a variant of self-supervised learning (Dosovitskiy et al., 2020; Devlin et al., 2018) that removes random input dimensions (also known as *masking*) in the observed data and learns a model that accurately predicts the missing values. This family of methods, well-known due to their success in natural language understanding, typically adopts a transformer architecture (Vaswani et al., 2017) as the feature extractor, that together with positional encodings and random masked dimensions allows capturing the bidirectional context in the data.

In BERT (Devlin et al., 2018), each sentence is usually considered as a $D$ dimensional observation vector, $\boldsymbol{x} = (x_1, x_2, \ldots, x_D)^\top$, where dimensions $x_t$ are named *tokens*. Given a *random mask* $\mathcal{M}$ of size $M < D$, as a set of indices drawn uniformly from $\{1, \ldots, D\}$, each token whose index belongs to $\mathcal{M}$ is considered to be in the subset $\boldsymbol{x}_\mathcal{M} = \{x_{\mathcal{M}(1)}, x_{\mathcal{M}(2)}, \ldots, x_{\mathcal{M}(M)}\}$. We refer to these as the *masked tokens*. The rest of indices $\mathcal{R} = \{1, 2, \ldots, D\} \setminus \mathcal{M}$ induce the complementary subset $\boldsymbol{x}_\mathcal{R}$, such that $\boldsymbol{x} = \boldsymbol{x}_\mathcal{M} \cup \boldsymbol{x}_\mathcal{R}$. Under this notation, MPT learns the parameters $\theta$ of a model $p_\theta(\cdot)$ by maximising an average of the following objective

$$\log p_\theta(\boldsymbol{x}_\mathcal{M} | \boldsymbol{x}_\mathcal{R}) \approx \sum_{t=1}^{M} \log p_\theta(x_{\mathcal{M}(t)} | \boldsymbol{x}_\mathcal{R}) \tag{1}$$

for every observation in the dataset $\mathcal{D}$. The stochastic choice of $\boldsymbol{x}_\mathcal{M}$ makes predictive conditionals $p_\theta(\boldsymbol{x}_\mathcal{M} | \boldsymbol{x}_\mathcal{R})$ to be different for every observation and training step. Once the pre-training of $\theta$ has converged, this naturally allows the model to capture the underlying structure between dimensions of the data. One additional remark is the number of random masks needed to cover all combinations between *masked* and observed tokens, which can be obtained as $\mathcal{C}_M = \binom{D}{M}$. In the particular example of BERT, where the masking rate is 15% with $D = 512$ and $M = 76$, the total number of random masks needed to cover all combinations of tokens is $\mathcal{C}_M \approx 1.21 \times 10^{92}$. This shows the inner combinatorial problem behind MPT. We provide empirical results on why this is not a limitation for learning with MPT in Sec. 3.1.

## 3 A probabilistic perspective, theory, and analysis

Our key objective is to demonstrate that the good generalization of MPT can be explained as an equivalence with the model's high marginal likelihood. Indeed, we will prove that MPT *implicitly* maximizes marginal likelihood according to some latent variable model of the form $p_\theta(\boldsymbol{x}|\boldsymbol{z})$.

**Marginal likelihood.** For our theory, we consider some dataset $\mathcal{D}$ consisting of $n$ i.i.d. observations $\boldsymbol{x}_{1:n}$, where each sample $\boldsymbol{x}_i$ could be either continuous or discrete and is of dimensionality $D$. We also assume that there exists a latent space $\mathcal{Z} \in \mathbb{R}^K$ where we can find unobserved variables $\boldsymbol{z}_{1:n}$ which are part of the generative process of the data. This assumption is inspired in the common use of latent encodings in recent models fitted with MPT. In this direction, we also consider the observations to be samples of a likelihood function $p_\theta(\boldsymbol{x}|\boldsymbol{z})$, where the mapping between the latent and observed variable is controlled by some parameters $\theta$, which might also include likelihood or prior *hyperparameters*.

Importantly, we consider the parameters $\theta$ to be *deterministic*, while we are interested in integrating out the latent variables that we cannot observe. Automatically, this leads us to the log-marginal

likelihood (LML) of the model, which may factorize as a sum of marginals and can be also written as $\log p_\theta(\boldsymbol{x}_{1:n}) = \sum_{i=1}^{n} \log p_\theta(\boldsymbol{x}_i)$, where the $i^{\text{th}}$ probability density comes from the integral $p_\theta(\boldsymbol{x}_i) = \int p_\theta(\boldsymbol{x}_i|\boldsymbol{z}_i)p(\boldsymbol{z}_i)\mathrm{d}\boldsymbol{z}_i$. This definition coincides with the target LML used in the lower bound of variational autoencoders (VAE) (Kingma and Welling, 2013; Rezende et al., 2014) and it is widely used in probabilistic generative models.

**Masking and conditional probabilities.** From the properties of probability distributions, we can decompose the individual LML functions $\log p_\theta(\boldsymbol{x}_i)$ as a sum of log-conditionals between *tokens*. Omitting the $i^{\text{th}}$ observation subscript in $\boldsymbol{x}$ to keep the notation uncluttered, the sum takes the form

$$\log p_\theta(\boldsymbol{x}) = \sum_{t=1}^{D} \log p_\theta\left(x_t|\boldsymbol{x}_{t+1:D}\right). \tag{2}$$

However, the previous sum imposes a particular order on the selection of variables for conditioning, e.g. $\{x_1|x_2, x_3, \dots\}, \{x_2|x_3, x_4, \dots\}$, etc. Moreover, the order of tokens in the observation vector remains predetermined, as dimensions are not *exchangeable*. Thus, we can consider a different combination of conditional probabilities in the sum — for instance, $\{x_4|x_1, x_2, \dots\}, \{x_3|x_1, x_2, \dots\}$, etc. Here, the key insight is that the rules of probability applied to the log-marginal likelihood make it *invariant* to the combination of different conditional factors, as we are observing different views of the same graphical model.

This combinatorial process between tokens in $\boldsymbol{x}$ can be understood as the selection problem of indices. For that reason, we can assume a mask $\mathcal{M}$ of the largest size $|\mathcal{M}| = D$, such that $\mathcal{M} \equiv \{1, 2, \cdots, D\}$. Using similar properties of combinatorics, we can also obtain $D!$ different choices for $\mathcal{M}$. While *all* the indices are always in the set, the order of indices differs between combinations. This principled order in $\mathcal{M}$ indicates how we sum the conditional probabilities in Eq. 2.

Since the LML is invariant to random choices of $\mathcal{M}$, we can re-write the sum in Eq. 2 as an expectation with a *countable set* of possible outcomes. Each outcome corresponds to one of the $D!$ choices for $\mathcal{M}$, such that

$$\log p_\theta(\boldsymbol{x}) = \frac{1}{D!} \sum_{\pi=1}^{D!} \sum_{t=1}^{D} \log p_\theta\left(x_{\mathcal{M}(t)}^{(\pi)}|\boldsymbol{x}_{\mathcal{M}(t+1:D)}^{(\pi)}\right) = \sum_{t=1}^{D} \mathbb{E}_\pi\left[\log p_\theta(x_{\mathcal{M}(t)}^{(\pi)}|\boldsymbol{x}_{\mathcal{M}(t+1:D)}^{(\pi)})\right], \tag{3}$$

where the superscript $(\pi)$ denotes which mask $\mathcal{M}$ are we using for indexing the tokens. We also *swapped* the order of the sums to obtain the desired expectation in the r.h.s. of the formula.

**The role of random masking.** If we now take a particular index $(t)$ and we look at the $\pi^{\text{th}}$ summand in the previous expression, we can see that the LML is still *invariant* to the order of the conditioning tokens $\boldsymbol{x}_{\mathcal{M}(t+1:D)}$ in the log-probabilities: $\log p_\theta(x_{\mathcal{M}(t)}|\boldsymbol{x}_{\mathcal{M}(t+1:D)})$ in the sum. Intuitively, we can use both — $\{x_1|x_2\}, \{x_2\}$ or $\{x_2|x_1\}, \{x_1\}$; independently of the conditional factors previously considered. In practice, this indicates that we can insert a *second set* of indices to the r.h.s. variables, which is the key point to link negative MPT loss and LML.

Now, assume that $\mathcal{M}$ indexes less than $100\%$ of tokens, while the rest is indexed by $\mathcal{R}$ as defined in Sec. 2. If we match both complementary masks to be aligned with the *conditional* and *conditioning* variables in the log-probabilities, this allows us to rewrite the $t^{\text{th}}$ summands in Eq. 2 as

$$\frac{1}{D!} \sum_{\pi=1}^{D!} \log p_\theta\left(x_{\mathcal{M}(t)}^{(\pi)}|\boldsymbol{x}_{\mathcal{M}(t+1:D)}^{(\pi)}\right) = \frac{1}{D!} \sum_{\pi=1}^{D!} \log p_\theta\left(x_{\mathcal{M}(t)}^{(\pi)}|\boldsymbol{x}_{\mathcal{R}(1:D-t)}^{(\pi)}\right).$$

Here, we can easily see that there are $\binom{D}{t-1}$ choices for the *unmasked* tokens in the r.h.s. of the conditional distribution, where we have previously fixed the index $t$. If we set the *binomial* coefficient $\mathcal{C}_t \equiv \binom{D}{t-1}$ as the maximum number of choices, we can obtain the following equality

$$\sum_{\pi=1}^{D!} \log p_\theta\left(x_{\mathcal{M}(t)}^{(\pi)}|\boldsymbol{x}_{\mathcal{R}(1:D-t)}^{(\pi)}\right) = \sum_{\pi=1}^{\mathcal{C}_t} \sum_{j=1}^{D-t+1} \log p_\theta\left(x_{\mathcal{M}(j)}^{(\pi)}|\boldsymbol{x}_{\mathcal{R}(1:D-t)}^{(\pi)}\right), \tag{4}$$

since $D! > \mathcal{C}_t \;\; \forall t \in \{1, 2, \dots, D\}$. Notice that once we have chosen a specific order $(\pi)$ in the masking pattern of $\mathcal{M}$ and $\mathcal{R}$ in Eq. 4, there are still $(D-t+1)$ choices for the *masked* tokens

under evaluation in the probability distribution. Alternatively, we can think of this method as taking advantage of the properties of probability to split the $D!$ choices in the order of log-conditionals into the two sums in Eq. 4. The driving idea is then that the two sums in the previous expression still remain *invariant* given any $t \in \{1, 2, \ldots, D\}$.

Using the previous notion in Eq. 2, we obtained our main result, which holds under the assumption of i.i.d. observations with correlated tokens and the previous definition of the LML as the integral over the stochastic latent variables in the model.

---

**Proposition 1** — *The cumulative expected loss of masked pre-training along the sizes of the mask of tokens $M \in \{1, 2, \ldots, D\}$ is equivalent to the log-marginal likelihood of the model when using self-predictive conditionals probabilities, such that*

$$\log p_\theta(\boldsymbol{x}) = \sum_{M=1}^{D} \mathcal{S}_\theta(\boldsymbol{x}; M), \tag{5}$$

*where the score function $\mathcal{S}_\theta(\cdot; M)$ corresponds to*

$$\mathcal{S}_\theta(\boldsymbol{x}; M) := \frac{1}{\mathcal{C}_M} \sum_{\pi=1}^{\mathcal{C}_M} \frac{1}{M} \sum_{j=1}^{M} \log p_\theta(x_{\mathcal{M}(j)}^{(\pi)} | \boldsymbol{x}_{\mathcal{R}(1:D-j)}^{(\pi)}) = \frac{1}{M} \mathbb{E}_\mathcal{M} \left[ \sum_{j=1}^{M} \log p_\theta(x_{\mathcal{M}(j)} | \boldsymbol{x}_\mathcal{R}) \right].$$

*Proof: In the supplementary material.*

---

It is remarkably important to link the sum of log-conditionals $\log p_\theta(x_{\mathcal{M}(j)} | \boldsymbol{x}_\mathcal{R})$ in our proposition with the main objective used in MPT in Eq. 1. The main message of our result is that the *score function $\mathcal{S}_\theta(\cdot; t)$ acts as an average* over the different random masks. These shape the structure of conditioning in probabilities. The cumulative sum of the score function $\mathcal{S}_\theta(\cdot; t)$ over different sizes of the MPT's mask formally leads to the true value of the model's LML. This result is exact whenever we consider the closed-form self-predictive probabilities of the model and *all* the possible choices for the masking pattern $\mathcal{M}$. Since this is usually not affordable, due to the combinatorial cost and the lack of tractability, we usually have a *biased* estimator. However, it is still sufficient to prove that MPT maximizes LML during training as we will show later. This point will be discussed in the following empirical studies. Further details on the derivations are included in the supplementary material.

### 3.1 Formal results in tractable models

To verify that masked pre-training effectively maximizes LML, we need a tractable probabilistic model based on latent variables as the *proof-of-concept*. Probabilistic PCA (PPCA) (Tipping and Bishop, 1999) is perhaps the best option here, as it has been previously used to understand other empirical observations in generative methods, e.g. posterior *collapse* in VAEs (Lucas et al., 2019), or even considered as the starting point of GPLVMs (Lawrence, 2005). In particular, the PPCA model assumes that Gaussian observations map linearly to sets of real-valued latent variables $\boldsymbol{z}_{1:n}$, such that $\boldsymbol{x} = \boldsymbol{W}\boldsymbol{z} + \boldsymbol{\mu} + \epsilon$, where $\epsilon \sim \mathcal{N}(0, \sigma_0^2 \mathbb{I})$. Importantly, the prior is conventionally defined as isotropic, where $p(\boldsymbol{z}) = \mathcal{N}(0, \boldsymbol{1})$. We are therefore interested in the closed form expression of the PPCA's LML, which also factorizes across samples as follows

$$\log p_\theta(\boldsymbol{x}_{1:n}) = \sum_{i=1}^{n} \log p_\theta(\boldsymbol{x}_i), \qquad \text{where} \quad p_\theta(\boldsymbol{x}_i) = \mathcal{N}(\boldsymbol{x}_i | \boldsymbol{\mu}, \boldsymbol{S}), \tag{6}$$

and we obtain the covariance matrix using $\boldsymbol{S} = \boldsymbol{W}\boldsymbol{W}^\top + \sigma_0^2 \mathbb{I}$. For our analysis, the Gaussian nature of $p_\theta(\boldsymbol{x}_i)$ is of fundamental importance. Given the random mask $\mathcal{M}$, the self-predictive conditionals used in MPT naturally emerge from the formulation using properties of Gaussian marginals, such that $p_\theta(\boldsymbol{x}_\mathcal{M} | \boldsymbol{x}_\mathcal{R}) = \mathcal{N}(\boldsymbol{m}_{\mathcal{M}|\mathcal{R}}, \boldsymbol{v}_{\mathcal{M}|\mathcal{R}})$ is parameterized according to

$$\boldsymbol{m}_{\mathcal{M}|\mathcal{R}} = \boldsymbol{S}_{\mathcal{M}\mathcal{R}}^\top \boldsymbol{S}_{\mathcal{R}\mathcal{R}}^{-1} \boldsymbol{x}_\mathcal{R}, \quad \boldsymbol{v}_{\mathcal{M}|\mathcal{R}} = \boldsymbol{S}_{\mathcal{M}\mathcal{M}} + \boldsymbol{S}_{\mathcal{M}\mathcal{R}}^\top \boldsymbol{S}_{\mathcal{R}\mathcal{R}}^{-1} \boldsymbol{S}_{\mathcal{M}\mathcal{R}}, \tag{7}$$

where we split the LML covariance matrix $\boldsymbol{S}$ into the blocks corresponding to the indices included in $\mathcal{M}$ and $\mathcal{R}$. We use these *mean* and *variance* parameters of the self-predictive density to recursively

evaluate the log-probabilities in Prop. 1. In practice, two elements become critical for the computation, one is the size $M$ of the *mask* and another is the number of random masks $P < \mathcal{C}_M$ considered. These induce a *trade-off* between accuracy and computational cost. Moreover, their role in approximating LML using a biased estimate is carefully analyzed in the following empirical studies. We also included additional details on the previous derivations in the supplementary material.

**Fast asymptotic convergence.** Our theory indicates that we should evaluate *all* $\mathcal{C}_M$ random masks of the tokens to achieve the exact value of the LML. However, even if the combinatorial nature of the sum in the r.h.s. of the last equation in Prop. 1 becomes very large when the dimensionality of data augments, we suspect that it might converge relatively fast to the true value of the LML. This hypothesis would explain why large models that are fitted with standard MPT generalize well using just one random mask per training epoch.

Here, we empirically study if the cumulative MPT loss converges to the *true* value of the LML under the definition of the PPCA model. In particular, to the LML obtained with the original parameters that generated the data. The results in Fig. 1 and Tab. 1 indicate that as long as we average over more random masking patterns, the cumulative MPT loss approximates the LML of the model very well. Thus, having defined a PPCA model with a latent space of $K = 2$ dimensions, we observe in the *left* and *middle* plots that the asymptotic convergence happens for both small ($D = 5$) and large ($D = 50$) number of tokens per observation. Additionally, we observe that the estimation of LML is clearly *unbiased* if we use the cumulative MPT loss according to Eq. 1, which is an important insight. Notice that $P = 1$ is usually set up in MPT in practice. Additionally, we tested the tractable model

Table 1: Evolution of negative MPT loss w.r.t. max. number of random masks $P$.

| TRUE LML ($\uparrow$) | $P = 1$ | $P = 10$ | $P = 100$ |
|---|---|---|---|
| $(-60.34)$ | $-60.44 \pm 0.47$ | $-60.22 \pm 0.12$ | $-60.34 \pm 0.03$ |

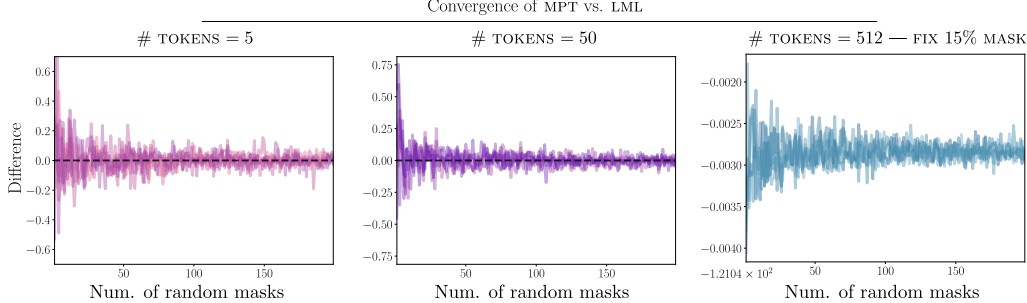

Figure 1: Asymptotic convergence of the cumulative MPT loss to LML as the number of random masks $P$ augments. Curves indicate the relative difference, where $0.0$ means that MPT equals LML. **(Left).** Each observation consists of 5 tokens. **(Center)** Each observation consists of 50 tokens. **(Right).** Observations have 512 tokens and the rate of masking is fixed to $15\%$ (76 tokens). As the theory indicates, when the size of $\mathcal{M}$ is fixed, the cumulative MPT loss becomes a *biased* estimator of the LML. The curves converge asymptotically to the bias.

using a dimensionality similar to the input data used in BERT (Devlin et al., 2018), where the number of tokens is typically $D = 512$ per observation and the mask rate is fixed to $15\%$. The fact of fixing the rate of masking in MPT produces that the sum in Eq. 5 is incomplete. Thus, we have a *biased* estimation of the LML. However, this bias is *known* and constant during the training of parameters $\theta$, which does not prevent the general maximization of LML. This point is carefully analyzed in the next empirical study with learning curves. One additional finding here is that as $P \to \mathcal{C}_M$, the cumulative MPT loss also converges asymptotically to the *biased* estimator of the LML as shown in the right plot in Fig. 1.

**LML maximization and biased estimation.** We next seek to extend the previous study to understand the behavior of the cumulative MPT loss in training curves. So far, we have observed how the number of random mask patterns affects the precision around the *unbiased* estimation of the

LML. Theory and previous empirical results indicate that we are targeting LML or at least a decent *biased* estimate of LML when averaging self-predictive conditionals as in MPT. However, we still want to examine if this maximizes LML in *all cases* and under stochastic gradient optimization. This principal hypothesis is confirmed in Fig. 2, where different training curves are shown for different initializations and setups of the same PPCA model. The key insight showed by this experiment is that the exact LML is iteratively maximized at each epoch, in parallel with the maximization of the negative MPT loss. On the other side, we also have that MPT is an unbiased stochastic approximation of LML, as in Fig. 1, whenever we consider different rates of random masking $\mathcal{M}$. We can also observe that as soon as we fix the size of the mask to index the 20% of tokens, the MPT loss becomes a *biased* estimate. Intuitively, this is equivalent to fixing $M$ in the sum in Eq. 5. Again, it converges to the same value from different initializations of parameters $\theta$. Additionally, we highlight that the LML is still maximized in this case, which is of high similarity to practical uses in larger models. Overall, this result first confirms the main insight of the work on the link between generalization when using MPT and the maximization of the model's LML.

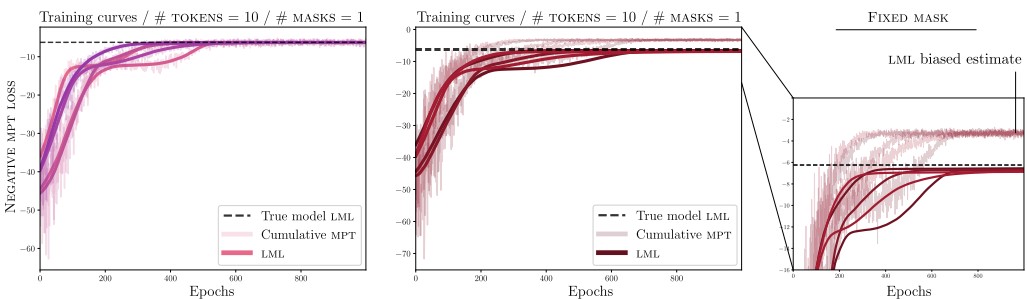

Figure 2: Training curves of the negative cumulative MPT loss in PPCA vs. the ground truth (GT) LML. The number of samples is $N = 2000$ and the number of tokens is $D = 10$. All plots used $P = 1$ random masks per epoch and five different initializations. **(Left).** The rate of masking is *unfixed* and it varies from 1% until 100%. The negative MPT loss converges to the GT-LML (dashed line). Darker curves are the exact LML per epoch. **(Center).** Convergence with *fixed* mask to 20% of tokens. The negative MPT loss is no longer centered around the LML and it converges to a *biased* estimate. **(Right).** *Zoomed* curves of convergence. The *bias* is constant and all MPT losses converge to the same point. The LML per epoch is also maximized and converges to GT-LML.

**Beyond tractable models and implicit integration.** One remaining question in our analysis is how the probabilistic theory around MPT adapts to intractable or non-linear models. In practice, self-predictive probabilities imply integrating out the latent variables, often given the posterior distribution. In most cases, performing this integration is extremely difficult or not possible in training time. Therefore, we are interested in finding if alternative approximations $q_\theta$ to the *true* self-conditional probabilities still produce accurate estimation and maximization of the LML. This point is confirmed in Fig. 3. Inspired by the experiments of Lucas et al. (2019) with *linear* VAEs, we set up a Bernoulli likelihood on top of the latent variable model. The tractable formulation in the Gaussian example coincides with PPCA. Since predictive conditionals are no longer tractable for us, we use numerical integration to obtain the probabilities of masked tokens. In Fig. 3, we test the training with the cumulative MPT loss as well as compare with standard variational inference using the model's evidence lower bound (ELBO). For the *mini*-dataset with MNIST samples, we observe that both models converge to a similar value of the LML. Thus, the fundamental insight here is that MPT maximizes LML even under training with approximate self-predictive conditional probabilities. For the LML curves, we also used numerical integration.

Beyond linear models, our theory is useful when applied to non-linear models. Moreover, in Fig. 3 we also include the results for *deep* VAEs based on NNs. While the estimation of LML was obtained via Monte Carlo (MC) samples, we used iterative *encoding-decoding* to produce the self-conditional probabilities for masked tokens — see Sec. F in Rezende et al. (2014). In this scenario, we also observe the maximization of the LML according to the evolution of the MPT loss.

Another key insight showed by this study is the ability of MPT to perform *implicit* integration. The cumulative sum over the different rates of random masking is another way to see a discrete integral

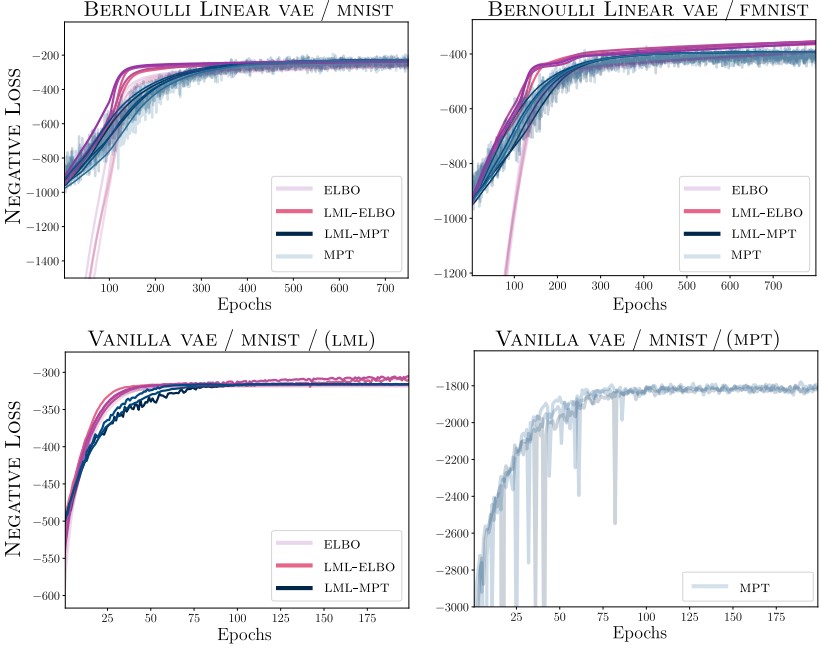

Figure 3: Training curves for linear VAE and deep VAE models with variational inference (VI) and MPT. Data consist of subsets of MNIST and FMNIST. **(Upper Row).** A linear VAE model with Bernoulli likelihood function in $N = 2000$ samples of MNIST and FMNIST. Shaded curves correspond to the target losses used in the optimizer (ELBO and MPT). Darker lines indicate the evolution of the LML, which are approximated via numerical integration in a latent space $\mathcal{Z}$ of dimensionality $K = 2$. **(Lower Row).** Vanilla VAE with Gaussian likelihood for MNIST. The LML curves are approximated via Monte Carlo (MC) samples. Self-predictive conditional probabilities are obtained via *recursive* encoding-decoding. The size of the random masking is fixed and set to 33%.

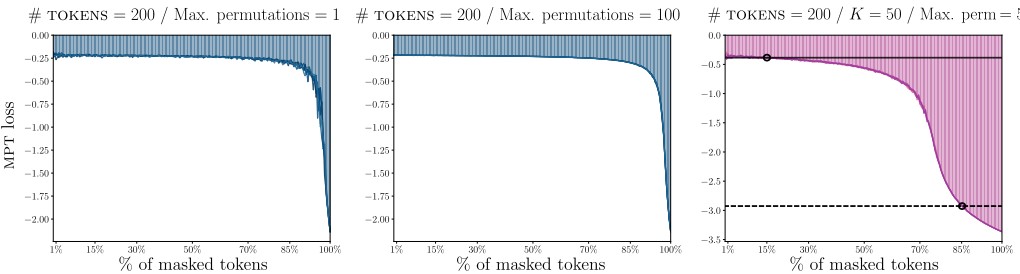

Figure 4: Area under the curve described by $\mathcal{S}_\theta(\cdot; M)$. The area is approximately equal to the model's LML according to the theory. Larger probability values are obtained with smaller rates of masking. **(Left).** Area described with $P = 1$ random masking per epoch. The curve is more noisy and the area slightly loses precision w.r.t. LML. **(Center). Area under the MPT curve for $P = 100$. (Right).** Latent space is augmented to be of $K = 50$. Decay of predictive probabilities begins at around 50% masking rate.

under the curve described by the score function $\mathcal{S}_\theta(\cdot; M)$ in Eq. 5. In Fig. 4, we show the areas under the curve and the effect of reducing the number of random masks $P$. The blue plots correspond to a trained PPCA model and the area corresponds to the LML estimate. The long tail in the right part of the curves, when the rate of masking is larger than 90%, indicates that the model is no longer able to produce good estimates of the tokens with only 10% of the input dimensions observed. This explains, why the probabilities have an approximately exponential decay. However, this effect is not constant, and it might depend on the latent structure of the model. In the r.h.s. plot we observe that the decay of conditional probabilities happens earlier at approximate 50% random masking or larger. The role of the masking rate is perhaps the missing part in the picture (Wettig et al., 2022), as it is the one that

determines the approximation to the LML. With the purpose of providing an intuition on how rates of 15% or 85% affect to the area under the curve, we indicate with two black lines the approximate area that approximates the LML. A longer discussion is provided in the supplementary material.

Table 2: Area under the MPT curve for BERT model and four GLUE datasets.

| GLUE datasets → | AX | COLA | QNLI | MRPC |
|---|---|---|---|---|
| Area / Random init. (↑) | −5245.31 | −5283.52 | −5343.98 | −5362.21 |
| Area / Pre-trained (↑) | −1715.75 | −1657.68 | −1770.28 | −1773.45 |

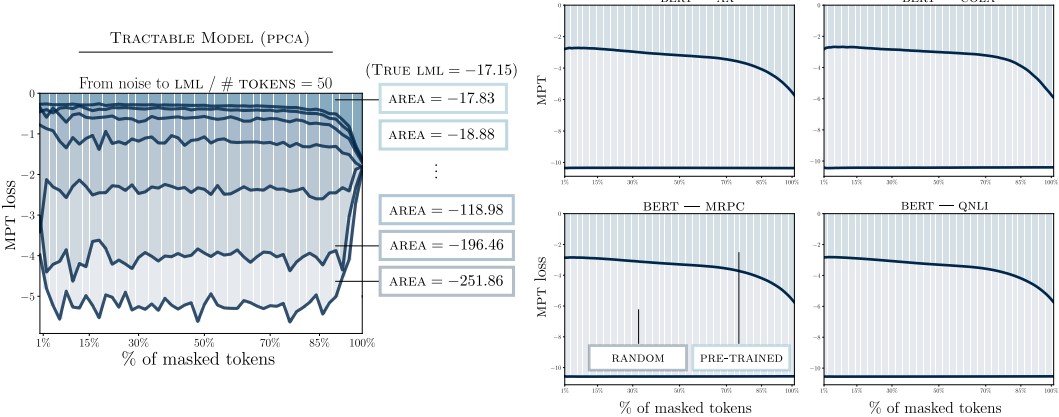

Figure 5: Evolution of the area under the MPT curve. Comparison between one tractable model (PPCA) and BERT. The area under the curves is *approximately* the LML. Random initialization of the parameters produces MPT curves with similar low probabilities for all % of masking. As the number of epochs increases, the curve brings higher values of log-probability for lower ratios of masking. The area also converges to the true value of LML. **(Left).** PPCA model trained for 600 epochs. Each curve represents {0, 100, 200, 300, 400, 500, 600} epochs of training with MPT. **(Right).** Random initialization and end-of-pretraining curves for the MPT loss w.r.t. the % of masked tokens. Curves are similar but not identical for the 4 different datasets given the pre-trained BERT model.

## 3.2 Applied theory on large language models

In this section, we aim to understand how the area under the MPT curve evolves and behaves for large language models (LLMs). While the direct computation of the LML is not feasible for non-linear transformer models, we are interested in checking how the rate of masking affects the curve compared with the tractable PPCA model. The results provided in Fig. 5 and Tab. 2 give us insights into this behavior. First, we observe that the MPT curve is approximately *flat* for every rate of masking in the PPCA when parameters are randomly initialized. Intuitively, this indicates that the model is not able to correctly predict any token given some context. In some way, it produces noise independently of the number of conditional tokens, which explains the low log-probabilities. Second, we can also notice that the curve changes its shape as more training epochs are considered. The curve after 600 epochs produces high probability values for different rates of masking, while the long tail of low probabilities appears when masking more than 85% of tokens. Moreover, the area under these curves is the estimation of the LML, which accurately converges to the *ground truth* value of the LML with the original generative parameters.

For the study of the curves in LLMs, we used four datasets from the General Language Understanding Evaluation (GLUE) (Wang et al., 2019). Additionally, we consider a 110M parameters BERT model using pre-trained checkpoints[2] and random initializations. To draw the MPT curves, we computed the mean cross-entropy per each rate of masking between 1% and 99%. In Fig. 5, we observe that random initializations of BERT parameters lead to *flat* curves of low self-predictive probabilities. On the other hand, the pre-trained curves show similar behavior as in the tractable model, where the area

---

[2]Pre-trained parameters for the BERT model are available in the library — `https://huggingface.co/`.

is reduced and a long tail of low probabilities happens when the rate of masking becomes larger. This result supports our hypothesis that MPT in LLMs might be performing implicit integration of the latent space and maximizing the marginal likelihood of the model.

**Results on vision models.** In addition to the results shown in Fig. 5 with the BERT model, we are also interested in the *behavior* of the theory on large models oriented to *vision*. For this study we provide the curves of the area under the MPT loss for VIT-MAE (Dosovitskiy et al., 2020; He et al., 2022) with different *masking* rates. In a similar way as in Sec. 3.2, we use an (already) pre-trained VIT-MAE model loaded from a public repository. To draw the curves shown in Fig. 6, we computed the losses per *each* rate of masking between 5% and 95% for samples from three different test image datasets (FASHION-MNIST, CIFAR-100 and TINY-IMAGENET). We can observe that the curves described by the pre-trained VIT-MAE model show a similar behaviour to the one we obtained with BERT and they are also *aligned* with the analysis done in this work with tractable models. We highlight that one must be aware that masking on vision models is often performed via *patches*. In practice, this could affect the expectation in Proposition 1, so this point should be taken into consideration if theory is applied to this case.

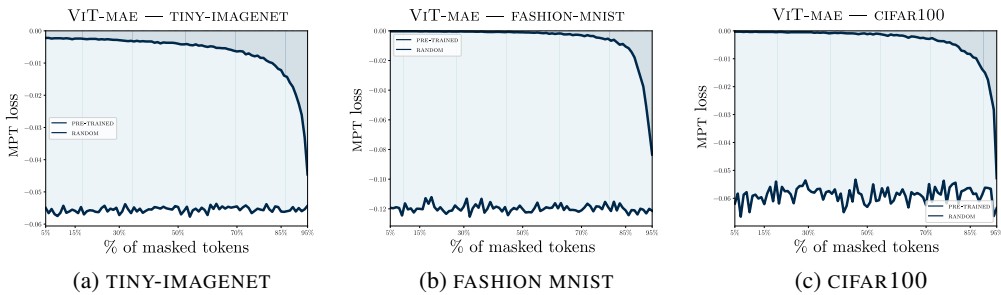

| (a) TINY-IMAGENET | (b) FASHION MNIST | (c) CIFAR100 |

Figure 6: Evolution of the area under the MPT curve. Comparison between three different datasets with VIT-MAE. The area under the curves is approximately the LML. Random initialization of the parameters produces MPT curves with similar low probabilities for all % of masking. As the number of epochs increases, the curve brings higher values of log-probability for lower ratios of masking. The area also converges to the true value of LML.

**Reproducibility.** All the empirical studies and results are *reproducible*. We provide the code and details for every figure in the public repository at `https://github.com/pmorenoz/MPT-LML/`.

## 4   Related work

Masked pre-training and large scalability are key elements of the current success of transformer models (Vaswani et al., 2017) on natural language processing (NLP) tasks. Vision transformers (VIT) (Dosovitskiy et al., 2020) bridged the architectural gap between NLP and computer vision, making masked language modeling (MLM) suitable for images. In this regard, BEIT (Bao et al., 2022) adopted VIT and proposed to mask and predict discrete visual tokens. Most recently, masked autoencoders (He et al., 2022) also adopted masked pre-training by predicting pixel values for each masked patch, and BEIT3 (Wang et al., 2022) performs MLM on texts, images, and image-text pairs, obtaining state-of-the-art performance on all-vision and vision-language tasks. Additionally, masked pre-training has been successfully adapted to video (Tong et al., 2022), where random temporal cubes are iteratively masked and reconstructed.

The surprising ability of recent generative models to generalize and do impressive in-context learning has inspired earlier works to study this phenomenon from the Bayesian lens. The notion that LLMs might be performing *implicit* Bayesian inference was first described in Xie et al. (2021) where in-context learning is described as a mixture of HMMs. However, the equivalence between the log-marginal likelihood and *exhaustive* cross-validation was first provided in Fong and Holmes (2020). Earlier works (Vehtari and Lampinen, 2002; Gelman et al., 2014) also provided a Bayesian perspective of CV. Additionally, Moreno-Muñoz et al. (2022) leveraged this link for training Gaussian process models according to a stochastic approximation to the marginal likelihood. Similarly to current masked pre-training, the size of the conditioning variable (masking rate) was held constant. This was reported to improve notably upon traditional variational lower bounds.

## 5 Discussion and outlook

In this paper, we have shown that masked pre-training implicitly performs stochastic maximization of the model's marginal likelihood. The latter is generally acknowledged as being an excellent measure of a model's ability to generalize (Fong and Holmes, 2020), and our results help to explain the strong empirical performance associated with masked pre-training. We have further seen that the developed theory matches the empirical training behavior well. Moreover, we illustrated the role that the rates and the number of random samples of masking play in the estimation of the LML. We have also provided insights and a new perspective to study masked pre-training in tractable models while also finding strong similarities with LLMs.

**Limitations.**   We have developed a formal probabilistic theory that links masked pre-training with the Bayesian principles. While we provide evidence that the impressive performance in recent large models is related to the maximization of the marginal likelihood, these methods usually introduce new elements of improvement but may not entirely fit the propositions provided in this work. In practice, this is not a limitation but a remark that there is still room for understanding the abilities of recent generative modeling. In this regard, one example might be autoregressive modeling between the masked tokens. While these are not currently analyzed in our work, we hypothesize that they could also be linked in further development to our formal propositions.

**Relevance for large models using masked pre-training.**   We have shown empirical results of the connection between MPT and LML. This link sheds light on the understanding of generalization, particularly in recent pre-trained models. One positive outcome of our studies is the notion of having *biased* Bayesian estimators whenever a practitioner fixes the masking rate, e.g. to $15\%$. Currently, there is a significant interest in the role of masking rates in LLMs (Wettig et al., 2022). These studies could benefit from the insights provided in this paper. We also argue that the theory offers *hints* that may be beneficial, for instance, for uniformly sampling the mask size, instead of the current fixed-rate practice. This practice is empirically shown in the supplementary material, and it leads to *unbiased* estimation which may result in better performance for certain scenarios (Tay et al., 2023).

**Relevance for Bayesian models.**   Current Bayesian modeling is dominated by approximate methods. Variational inference foregoes the ambition of training according to the marginal likelihood and instead resorts to bounds thereof. This inherently yields suboptimal models. Our theory suggests that if we can design Bayesian models in which conditioning is *cheap*, then we can stochastically optimize w.r.t. the true marginal likelihood easily. Beyond shedding light on the success of masked pre-training, the theory also suggests that large-scale Bayesian models could be successfully trained in the future with appropriately designed self-supervision.

## Acknowledgements

The authors want to thank Yingzhen Li for her inspiring talk on MPT during the GENU'22 meeting in Copenhagen. For us, this was the starting point of fruitful discussions that led to this final work. This project has received funding from the European Research Council (ERC) under the European Union's Horizon 2020 research and innovation programme (grant agreement 757360). This work was in part funded in part by the Novo Nordisk Foundation through the Center for Basic Machine Learning Research in Life Science (NNF20OC0062606). SH was also supported in part by a research grant (42062) from VILLUM FONDEN.

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
