# OpenReview forum: "On Masked Pre-training and the Marginal Likelihood"
_NeurIPS.cc/2023/Conference — NeurIPS 2023 poster_

### Official Review · Reviewer_vzMk · 2023-07-05

**Soundness:** 3 good
**Presentation:** 3 good
**Contribution:** 2 fair
**Rating:** 4
**Confidence:** 4

**Summary:**

The authors set out to show that the good generalization of Masked Pre-Training can be explained as the equivalence with model's high marginal likelihood. The exchangeability assumption in sequential inputs is handled via a combinatorial choice over the subset of masking features, such that the masked pre-training objectives is a marginalization over uniform distribution of different maskings. The rest of the paper is a discussion on the implications of such a formulation.

**Strengths:**

The paper is reasonably written, and the authors have made a decent effort to convey the significance of the work.

**Weaknesses:**

I am not very convinced by the arguments in the paper that claim that maximizing the marginal likelihood is the reason behind generalization of MPT.

In Line 182, the authors claim an "insight" that MPT targets a biased estimate of LML by showing that fixing the masking rate to 20% of  tokens, MPT becomes biased and converges to a different value. I feel like this argument could be made for any two objectives, and something remains missing in this justification. From the reverse-engineered theory, it is already obvious that MPT introduces bias and positioning this practical result as an insight seems to not add any new information. I'd appreciate a clarification here.

In Line 212, the authors claim that MPT performs implicit integration. However, this claim also seems a little unfounded. When averaging over the % of masked tokens in Figure 4, if bulk of the integrated mass is constant, then an alternate way to approximate the integral would be to just take the constant mass (since the integral contains averaging terms as well). Doesn't this result confound the conclusion and just a matter of coincidence?

In any case, marginal likelihood just answers a question very different from generalization - marginal likelihood tells us the likelihood of the data under the "prior" model, i.e. how well the priors explain the data. It does not have anything to do with what happens to the model after training, in principle. There could be confounders to the "insights" claimed by the authors here, especially with large models, since the choice of number of tokens, number of maskings, are far-off from what model large language models use.

**Questions:**

1. Perhaps I am misunderstanding, but shouldn't Eq. (2) be conditioned on all feature variables $1:D$ except for the one being predicted? Eq. (3) seems to present is correctly, since it clearly distinguishes masking indices.

**Limitations:**

Yes. See my comments on weaknesses and questions.

---

> ### Author Rebuttal · Authors · 2023-08-09
>
> Thanks for the constructive feedback and the time taken to review our paper. We have addressed all your comments individually below. If any concerns remain, we would be also happy to clarify.
>
> **Point 1** & **Point 2**
> > *I am not very convinced by the arguments in the paper that claim that maximizing the marginal likelihood is the reason behind generalization of MPT.*
>
> > *In any case, marginal likelihood just answers a question very different from generalization - (...)*
>
> We would like to remark that this is not the exact message that we want to send in our paper. Precisely, we are interested in the link between MPT and the log-marginal likelihood. Thus, the main discovery that we show is that log-marginal likelihood is equivalent to the cumulative sum of MPT losses shaped with different sizes of the mask. The final conclusion of our work is that MPT implicitly performs stochastic maximisation of the model's marginal likelihood. That is what we prove in a rigorous manner. Additionally, these points are confirmed with the empirical results provided on different tractable and intractable models.
>
> The fact that MPT implicitly maximises the model's marginal likelihood is important. Indeed, the marginal likelihood (or evidence) has been longly considered as the measure of generalization ability in Bayesian models. As we are sure the reviewer knows, LML is usually the desired loss that one would like to obtain in Bayesian modelling, with a large list of examples where Laplace approximations for Bayesian NNs recently excell. However, this one is usually difficult to compute due to the prohibitive computational cost or intractability in the integrals.
>
> Hence, having this connection is perhaps not the final answer to explain why MPT works so well in the recent advances, but we believe that the empirical evidence is a step forward to explain at least one of the causes why pre-trained models generalize so well.
>
> **Point 3**
> > *In Line 182, the authors claim an "insight" that MPT targets a biased estimate of LML by showing that fixing the masking rate to 20% of tokens, (...).*
>
> Fair point. So far, before the section that begins in L175, we have proven and provided empirical results which indicate that the cumulative MPT losses with a masking rate from 1% to 100% equals the LML of the model. However, we know that this is not what MPT does in practice, where the masking rate is usually fixed.
>
> We were particularly interested in the understanding the effect of fixing the masking rate. Perhaps, the main surprise is not having a biased estimate of the LML probabilities, but that the biased is **fixed** with respect to the learnable parameters. This is not a common thing, and having biased estimation is a usually a huge problem. However, in the results that we provide this bias is not negatively affecting the maximisation of the LML as it is indicated in Figures 2,3 and 5.
>
> On the other hand, we were interested in understanding the meaning of setting a fixed masking rate of  20, 30, 40 or even 50%. For that reason, we included additional empirical results in Figure 4, where the $x$-axis is the % of masked tokens. Notice that the area under the curve is the LML, which is maximised as the MPT losses decrease.
>
> **Point 4**
> > *In Line 212, the authors claim that MPT performs implicit integration. However, this claim also seems a little unfounded. (...)*
>
> We understand the concern here, but we do not think that the claim on *implicit integration* is *unfounded*. We remark that the **expectation integral** plays a key role in the main equation in Proposition 1. Moreover, as the reviewer correctly points out, we focus on the area under the curve described by the MPT losses with respect to the % of masking rate. These curves can be seen in Figures 4 and 5.
>
> The area under the curve is not necessarily constant, and fixing the masking rate basically sets an approximation based on the constant mass. Depending on which decisions are taken, the integration of the area is performed way with one approximation or another. But importantly, none of this is done on purpose, but *implicitly* when we average losses in MPT. We do not think that all of this is just a **coincidence**, as theory goes first and empirical results later. In that sense, we encourage the reviewer to check the left-hand plot in Figure 5, where the area under the curve converges to the true value of the model's LML.
>
> **Point 5**
> > *Perhaps I am misunderstanding, but shouldn't Eq. (2) be conditioned on all feature variables $1:D$ except for the one being predicted? (...)*
>
> In particular, Equation (2) uses rules of probability (i.e. chain rule) to make the factorisation of $\log p_{\theta}(\boldsymbol{x})$. Notice that the object $\boldsymbol{x}$ included all the observed variables $x_1, x_2, x_3, \dots, x_D$. In this way, the density $\log p_{\theta}(\boldsymbol{x})$ works as a joint distribution $\log p_{\theta}(x_1, x_2, \dots, x_D)$. If we apply one time the chain rule of conditional probability to this joint distribution, we easily get the following factorisation
>
> $$\log p_{\theta}(x_1, x_2, \dots, x_D) = \log p_{\theta}(x_1| x_2, \dots, x_D) +\log p_{\theta}(x_2, \dots, x_D).$$
>
> Doing it again on the right hand side log-density $\log p_{\theta}(x_2, \dots, x_D)$ gives us again an extra conditional term to the sum, such that
>
> $$\log p_{\theta}(x_1, x_2, \dots, x_D) = \log p_{\theta}(x_1| x_2, \dots, x_D) +\log p_{\theta}(x_2 | x_3, \dots, x_D) + \log p_{\theta}(x_3, \dots, x_D).$$
>
> If we recursively apply this property, we obtain the sum expressed in Eq. (2), which does not requires to have distributions conditioned on all the feature variables $1:D$. We hope that this clarifies a bit this point, as it is a key property in the rest of derivations used for the proof.

---

> > ### Comment · Reviewer_vzMk · 2023-08-14
> >
> > >  Indeed, the marginal likelihood (or evidence) has been longly considered as the measure of generalization ability in Bayesian models.
> >
> > I think this statement has much more nuance to it than how it is often directly inherited from early works of David MacKay. In many cases, it comes out often objectively wrong too, e.g. see [1].
> >
> > > The fact that MPT implicitly maximises the model's marginal likelihood is important.
> >
> > All details aside, I think this would be the key disagreement I have, and the main hesitation to leaning towards an accept. The connection between MPT and optimizing LML is claimed to be "important". The paper, however, spends a lot of time making and supporting the connection, but not on why this connection is "important". And the fact that other supporting literature that relies on LML has not quite made enough of a mark, makes me less confident about the importance of such a connection, if at all it is practically relevant.
> >
> > Of course, I empathize with the point that this is only the first step, and there are serious computational challenges.
> >
> > [1]: https://proceedings.mlr.press/v162/lotfi22a.html
> >
> > ---
> >
> > Thank you for all the other clarifications as well. I'll keep my score to reflect my confidence in the connection's ability to influence practical applications, given that you have enough support from other reviewers. :-)

---

> > > ### Author Response · Authors · 2023-08-15
> > > **Response to additional comments**
> > >
> > > Dear reviewer, thanks for the time spent to read our rebuttal and the additional feedback provided. We would like to add some comments to your main points.
> > >
> > > > *I think this statement has much more nuance to it than how it is often directly inherited from early works of David MacKay. In many cases, it comes out often objectively wrong too, e.g. see Lotfi et al. ICML 2022*
> > >
> > > 1. We appreciate the reference attached, which is one recent paper that argues in the context of Bayesian NNs that LML might be misaligned with generalisation (in some cases). However, this paper is one against a whole thread of previous works that provide empirical results on the opposite direction, e.g. from all the inherited works since David MacKay, including foundational probabilistic approaches from Z. Ghahramani until the main principles behind the success of Gaussian processes.
> > >
> > > 	We must say that the skepticism around LML and its role in Bayesian NNs is a hot-topic in the community right now, which mainly comes from the difficulties faced in the last years, for instance, the posterior cold effect, the proper priors for weights and prohibitive computational costs.
> > >
> > > 2. In general we think that saying that LML hasn't had a practical value so far, thereby disregarding decades of work in the Bayesian community is perhaps not a good idea.
> > >
> > > 	Nonetheless, the reviewer is right in the sense that LML has not yet made a practical difference in deep learning, but it does not mean much as nobody currently knows how to even evaluate the LML with very-large NNs. That is, complete relevant experiments in that direction have not been yet conducted, unfortunately.
> > >
> > > > *And the fact that other supporting literature that relies on LML has not quite made enough of a mark, makes me less confident about the importance of such a connection, if at all it is practically relevant.*
> > >
> > > 3. Additionally, the argument that evaluating the LML is not practically relevant is a *chicken-and-egg* type of problem: the Bayesian hypothesis regarding LML cannot be (yet) properly tested without tools to evaluate the LML, and tools for evaluating the LML cannot be move forward because the LML may not be valuable.
> > >
> > > > *All details aside, I think this would be the key disagreement I have, and the main hesitation to leaning towards an accept.*
> > >
> > > > *I'll keep my score to reflect my confidence in the connection's ability to influence practical applications, given that you have enough support from other reviewers. :-)*
> > >
> > > We understand the skepticism around the previous points on the LML, but in our opinion, such debate is perhaps out of the scope of our work and on a different direction. We remark that we were interested in building a link between the LML and masked pre-training. Thus, we hoped it to be a first step towards understanding and a first door open to all the Bayesian literature to make contributions for the immensely large effort needed in the new MPT techniques and novel models as LLMs. But not to make all the new advances to be led by LML or the Bayesian paradigm, that's a different matter.
> > >
> > > Finally, we hope that you could re-consider the hesitation to leaning towards an acceptance, and in such case, if some concerns remain we are of course happy to answer. :)
> > >
> > > ```
> > > S. Lotfi, P. Izmailov, G. Benton, M. Goldblum and A. G. Wilson, Bayesian Model Selection, the Marginal Likelihood, and Generalization, ICML 2020
> > > ```

---

> > > > ### Comment · Reviewer_vzMk · 2023-08-16
> > > >
> > > > > In general we think that saying that LML hasn't had a practical value so far,
> > > >
> > > > I agree, I made a broader statement than I intended to. I meant LML hasn't had practical value so far *in the context of* deep neural networks. GPs have been the flag bearers of LML, of course.
> > > >
> > > > > Additionally, the argument that evaluating the LML is not practically relevant is a chicken-and-egg type of problem: the Bayesian hypothesis regarding LML cannot be (yet) properly tested without tools to evaluate the LML, and tools for evaluating the LML cannot be move forward because the LML may not be valuable.
> > > >
> > > > Let me be clear though, I don't intend to shoot down the possibility of LML being effective at all. Even the paper I referenced does not claim that. It would imprudent to make such a blanket statement.
> > > >
> > > > > we hoped it to be a first step towards understanding and a first door open to all the Bayesian literature
> > > >
> > > > What makes me uneasy, however, is the fact that a claim that "here's a connection" without any demonstration of the implications of such a connection is incomplete.
> > > >
> > > > Of course, your work is not the first one to do this. There is plenty of published work which makes the connection without a demonstration of a single demonstration of even "why", which, arguably is a weaker requirement than "how". And subsequently, the connections do not lead to any new development either.
> > > >
> > > > > But not to make all the new advances to be led by LML or the Bayesian paradigm, that's a different matter.
> > > >
> > > > We'll keep running in circles with this argument. To the make the case easier for you (and the ACs when they read this), I am certainly sympathetic to your argument that such a demonstration is hard with the current tools we have for NNs.

---

> > > > > ### Author Response · Authors · 2023-08-18
> > > > >
> > > > > Once again, we thank the reviewer for keeping this discussion alive.
> > > > >
> > > > > > *I agree, I made a broader statement than I intended to. (...)*
> > > > >
> > > > > > *Let me be clear though, I don't intend to shoot down the possibility of LML being effective at all. (...)*
> > > > >
> > > > > We appreciate that the reviewer acknowledges that some previous statements could be too broad in the sense of the practical utility of LML. We agree that there is still plenty of work yet to test LML in deep NNs, but we consider such to be out of scope for the present paper.
> > > > >
> > > > > > *There is plenty of published work which makes the connection without a demonstration of a single demonstration of even "why", which, arguably is a weaker requirement than "how" (...)*
> > > > >
> > > > > We understand the point that the reviewer intends to show. In some way, we think that it is kind of a skepticism around the potential utility of LML in the big picture drawn by the a last advances with large language models. However, this opinion is perhaps too broad to what our work is saying about the utility of LML in the context of MPT. Additionally, we mainly think of the purpose of the paper as being about allowing probabilistic models to be trained using self-supervision (once the link is established and MPT is understood), but it's also fair if you view the purpose differently.
> > > > >
> > > > > We also think that requiring the described big "why" as the reason to finally lean towards acceptance is perhaps a prohibitive milestone for us. As the reviewer also indicated, there is plenty of published works that are not able to provide that answer.
> > > > >
> > > > > > (...) *is the fact that a claim that "here's a connection" without any demonstration of the implications of such a connection is incomplete.*
> > > > >
> > > > > We quickly remind the structure of the paper, where we built the connection between MPT and LML, being the latter used as the common measure of generalization ability in Bayesian models. Additionally, we provided plenty of empirical results that support the theoretical link and are fully reproducible. Moreover, we included new clarifications and additional experiments in this rebuttal. Is not all of this enough?
> > > > >
> > > > > > *We'll keep running in circles with this argument (...)*
> > > > >
> > > > > Overall, we find that these subjective debates (that often run in circles) are what makes NeurIPS a great conference to attend and to discuss in person. For that reason, we kindly ask you to re-consider your hesitation, as it would be great if the paper was accepted such that more could join the debate.

---

### Official Review · Reviewer_LKWQ · 2023-07-06

**Soundness:** 3 good
**Presentation:** 3 good
**Contribution:** 3 good
**Rating:** 6
**Confidence:** 2

**Summary:**

This paper derives the equivalence between log marginal likelihood and negative masked pre-training loss which is often used for training text models. The paper argues that as marginal likelihood is the Bayesian way to do model selection, doing training using the masked pre-training loss inherits good generalization properties.

Experiments are done on a probabilistic PCA model, VAE and BERT. In PPCA, where the ground truth is known and the equivalence is confirmed empirically. On this model, it is empirically confirmed that even when we optimize a biased estimation of the log marginal likelihood, different training runs still converge to the same log marginal likelihood, with a slight offset to the true value.

In VAE, we don't know the ground truth, but verify that optimizing using the masked pretraining loss is similar to optimizing with ELBO, in terms of the log marginal likelihood value that the model converges to.

In the BERT model, we simply observe that the shape of the S(x, M) term is similar for PPCA and BERT during different points of training.

**Strengths:**

- This is an important connection to make and the authors discuss relevance for LLMs and Bayesian learning in the discussion.
- The experiments study the realistic case in which we can't enumerate over all masks and verify that masked pre-training is still similar to log marginal likelihood training

**Weaknesses:**

- The derivations are likely correct but I couldn't establish this 100%, see questions below.
- Isn't the Transformer just trained with the autoregressive log likelihood (and not with masked pre-training)? And isn't ViT simply trained to predict object class -- where is the masking?
- The jump to Proposition 1 was big for me and the preceding intuition building parts didn't help

**Questions:**

- In equation (1), do we not condition on $\mathbf x_{\mathcal M \setminus \{\mathcal M(t)\}}$ -- how is the marginalization over this variable done in language models?
- Is there a reason why we condition on $\mathbf x_{t + 1:D}$ instead of $\mathbf x_{1:t}$?
- In equation before equation 4, we assume $M < D$, so the value $\mathbf x_D$ which is part of $\mathbf x_{\mathcal M(t+ 1:D)}$ in the left hand side doesn't exist
- In line 109, shouldn't it be ${D \choose t}$ instead of ${D \choose t - 1}$ since there are $t$ tokens we condition on?


**Limitations:**

-

---

> ### Author Rebuttal · Authors · 2023-08-09
>
> We thank the reviewer for the positive consideration of our work, and the useful feedback including the technical questions on the derivations. We are interested in addressing all the questions brought in the review for a full understanding of the reviewer. If there is still something unclear, we would be happy to provide extra responses.
>
> **Point 1**
> > *Isn't the Transformer just trained with the autoregressive log likelihood (and not with masked pre-training)? And isn't ViT simply trained to predict object class -- where is the masking?*
>
> Fair point. As the reviewer indicates, Transformers and ViT architectures themselves don't use masked pre-training (MPT). But we do not mention Transformer or ViT architectures as the models considered in our analysis of MPT. Neither on the related work section. However, BERT and MAE models do use Transformers and ViT as the backbone method respectively while, at the same time, they use MPT to generate the self-supervisory signals. Precisely, this pre-training methodology is the point were we focus our theoretical analysis and connection with LML.
>
> Regarding ViT and vision models, we have now included new empirical results with the ViT-MAE model. These are shown in the PDF attached in the global response to all reviewers.
>
> **Point 2**
> > *In equation (1), do we not condition on $x_{\mathcal{M}/\mathcal{M}(t)}$ -- how is the marginalisation over this variable done in language models?*
>
> Assuming you refer to the masked tokens $x_{\mathcal{M}(t)}$, we do not condition on them as they are the ones we want to predict. These ones are indeed the self-supervisory signals that makes the model to learn a holistic structure of the data. Notice that we are recurrently conditioning on the rest of tokens $x_{\mathcal{R}}$. Importantly, the observed data is never marginalised and the log-marginal likelihood (LML) builds a probability metric over all the tokens.
>
> **Point 3**
> > *Is there a reason why we condition on $x_{t+1:D}$ instead of $x_{t+1}$?*
>
> If we understand correctly, the reviewer is referring to Equation (2), were we apply properties of conditional probabilities to factorise the $\log p_{\theta}(x)$ distribution or LML. We highlight  that in the object $x$ we have $D$ variables $x_1,x_2,\dots, x_D$. Thus, the goal of the sum is to obtain a recursive summation of uni-variate conditional distributions, for instance, $\log p_{\theta}(x_1|x_2,x_3, \dots ,x_D) + \log p_{\theta}(x_2|x_3, \dots ,x_D) + \dots$ That's why we use $x_{t+1:D}$ instead of $x_{t+1}$ in the conditioning variables.
>
> **Point 4**
> > *In equation before equation 4, we assume $M < D$, so the value $x_D$ which is part of $x_{\mathcal{M}(t+1:D)}$ in the left hand side doesn't exist*
>
> Right, maybe this was not clear enough. The equation indicated by the reviewer says that we have multiple choices for the order of conditioning objects in the right hand side of log probabilities. Importantly, the letter $\mathcal{M}$ indicates the indices, such that $\mathcal{M} = \{1,2,\dots,D\}$ as is described in L94.
>
> The key point in this equation is that we are fixing the index $t$, where $1<t<D$ such that we can analyse with all the mentioned combinations for $x^{(\pi)}_{\mathcal{M}(t+1:D)}$  Thus, the tokens included in the list of indices $\mathcal{M}(t+1:D)$ depend on the choice $\pi$ and there is not really a problem with the variable $x_D$.
>
> We highlight that the variable $M$ (size of mask) is different from the variable $\mathcal{M}$ (indices of tokens), which could be the cause of a confusion here. We hope that this clarifies this point and we will do our best to update the section on this regard.
>
> **Point 5**
> > *In line 109, shouldn't it be $\binom{D}{t}$ instead of $\binom{D}{t-1}$ since there are tokens we condition on?*
>
> Fair point. This critical point is perhaps a bit counterintuitive. We will update it in a way that is clearer. The key idea in L109 is that, fixing an index $t$, we have $D-t+1$ tokens to predict probabilities on (left hand side variables $x_{\mathcal{M}(t)}$ in the conditional distribution), and D-choose-(t-1) distinct orders for the conditioning tokens $x^{(\pi)}_{\mathcal{M}(t+1:D)}$ (*unmasked*). That is, we represent the latter choices with the binomial coefficient $\mathcal{C_t} = \binom{C}{t-1}$. Extra details on these derivations are also included in the Appendix A of the paper, where all elements are placed together for a better understanding.

---

### Official Review · Reviewer_Gnha · 2023-07-06

**Soundness:** 4 excellent
**Presentation:** 3 good
**Contribution:** 3 good
**Rating:** 7
**Confidence:** 2

**Summary:**

This paper shows that MLM is effectively maximizing the model's marginal likelihood, perhaps explaining why MLM has been successful. Beyond providing a proof, they run several empirical experiments suggesting their results hold in practice.

**Strengths:**

1. Clear introduction; Clear highlighting of paper strengths.
2. Clear background on MLM.
3. They provide a proof in the appendix and a number of empirical tests that support their findings in practice.
4. Their discussion showing connections both to model design in AI as well as relevance to improving Bayesian Models seems well put together.

I don't deeply understand the proof and math, so am I unable to fully judge the strength of the contribution. For what I do appreciate, I would recommend the paper to be accepted. I found the paper to be interesting, overall well-presented, and noteworthy.

**Weaknesses:**

1. Providing more hand-holding between the various subsections of section 3 would help the reader understand your goals for each section. Right now I can see the specific finding, but: Stressing the greater point that you're trying to verify your proof in practice (?) in each of these subsections and where and how you find support would be great.
2. Providing slightly more hand-holding on what marginal likelihood is and what it would be capturing in the context of, say, BERT, would make the paper stronger.

Nits
1. Figure 1: The y-axis get smaller (key detail!) but I could not see this initially. The text was very light. Perhaps increasing the font size somehow or making more of a note of it in the caption.
2. Figure 5: y-axis is both small and cut off.

**Questions:**

On line 207 (and beyond) you make some distinction between linear vs non-linear models. Does your proof make assumptions along these lines? If so, can you bring these assumptions further up into the abstract?

**Limitations:**

"The proofs can either appear in the main paper or the supplemental material, but if they appear in the supplemental material, authors are encouraged to provide a short proof sketch to provide intuition." More of a proof sketch might help the reader understand the footing of the paper.

---

> ### Author Rebuttal · Authors · 2023-08-09
>
> We thank the reviewer for the positive consideration and the useful feedback on our work. We also appreciate the highlighted strengths and contributions. We addressed the main comments and concerns included in your review. If any other concern remains, we would be also happy to clarify during the discussion period.
>
> **Point 1 & 2**
> > *Providing more hand-holding between the various subsections of section 3 would help (...)*
>
> > *Providing slightly more hand-holding on what marginal likelihood is and what it would be capturing in the context of, say, BERT, would make the paper stronger.*
>
> We appreciate the insightful feedback for making the paper *clearer* and *stronger*. At the beginning, our main concern was on the structure of the paper. Mainly, we had to choose between the two views described in the end of **Section 5**. That is, writing the paper to highlight the relevance for Bayesian ML practitioners (who might not be aware of the role that MPT plays), or for the audience who knows MPT very well and would like to use the proposed link for a better understanding. Perhaps, we implicitly chose the first view, whose main consequence are the two points mentioned by the reviewer. We will update it accordingly to your comments if the paper goes on for a final version.
>
> **Point 3**
> > *On line 207 (and beyond) you make some distinction between linear vs non-linear models. Does your proof make assumptions along these lines? (...)*
>
> Fair point. The paper is organised in a way where we first focus on the connection between masked pre-training (MPT) and log-marginal likelihood (LML), independently of the type of probabilistic model chosen (e.g. linear as PPCA or non-linear as VAEs). Importantly, the proof we provide **does not make any assumption** along the lines on the linear and non-linear nature of the probabilistic model. This is an important point that we would like to emphasize.
>
> Once we provide the main theoretical results for the connection between MPT and LML, we jumped into the second part, where our interest was providing new empirical results to the previous theory. Since for linear models the calculus of the LML is generally **exact**, we used PPCA as the first probabilistic model for verification. This is inspired by Lucas et al., NeurIPS 2019, who also used PPCA for analysing the behavior of VAEs and particularly, the *posterior collapse* effect.
>
> The sentence on L207 makes reference to the applicability of the theory (i.e. connection between MPT and LML) beyond tractable linear models, that is, non-linear models. We address this point in the following sections with empirical results with VAEs and LLMs (i.e. BERT).
>
> ```
> J. Lucas, G. Tucker, R. Grosse and M. Norouzi. *Don’t Blame the ELBO! A Linear VAE Perspective on Posterior Collapse*. NeurIPS 2019
> ```
>
> **Point 4**
> > *"The proofs can either appear in the main paper or the supplemental material, but if they appear in the supplemental material, authors are encouraged to provide a short proof sketch to provide intuition." More of a proof sketch might help the reader understand the footing of the paper.*
>
> We appreciate your recommendation. Our first idea was to include the main proof in the paper, however, we chose an intermediate solution which can be checked between L103-L130, including the **Proposition 1**. We shaped this as a proof sketch, but considering your feedback, we would be happy to restructure this part and include extra details if the paper moves forward.

---

> > ### Comment · Reviewer_Gnha · 2023-08-18
> >
> > Thanks for the helpful response!
> >
> > I still recommend an accept but with low confidence because I feel I don't quite understand some of the proof nor some of the issues the other reviewers are pointing out.

---

> > > ### Author Response · Authors · 2023-08-19
> > >
> > > Thank you for the support. If there's anything you would like to discuss or clarify then we are more than happy to discuss further.

---

### Official Review · Reviewer_mVT4 · 2023-07-07

**Soundness:** 3 good
**Presentation:** 3 good
**Contribution:** 2 fair
**Rating:** 4
**Confidence:** 4

**Summary:**

They show empirically that randomly masking a fixed number of M tokens and predicting with the remaining tokens produces a biased estimate of the log marginal likelihood $\log p(x)$ (LML).
Furthermore, they prove that repeating the masking for M from 1 to D (the total number of tokens), and summing up the estimates, leads to an unbiased estimate of the LML.

**Strengths:**

- The paper is well written and mostly easy to follow.
- Intuitive proofs and arguments are well supported by empirical results for tractable and intractable models.
- The work might give insights/ideas for setting masking rates in pre-training.


**Weaknesses:**

Main Concern:
The main result Proposition 1, requires to condition on different subsets of the remaining the tokens $R$. But this is not what is done in practice for masked pre-training, where normally all tokens in $R$ are used. It would have been interesting to see some discussion/experiments about the effect.

Others:
- It would have been interesting to see some experiments about the ideas stated in line 295.
- Given the works like Fong and Holmes (2020), there is no so much novelty from a theoretical point of view.

**Questions:**

- Could you comment on my main concern above?
- Why does the LML change with the number of epochs?
- What is GT-LML?

**Limitations:**

see main concern.

---

> ### Author Rebuttal · Authors · 2023-08-09
>
> We thank the reviewer for the acknowledgement of the main contributions of our work, the useful comments and the relevant feedback provided on the technical side. We have addressed _all_ your comments individually in the lines below.  If there is still something unclear, we would be also happy to clarify with extra responses.
>
> **Point 1**
> > *The main result Proposition 1, requires to condition on different subsets of the remaining the tokens $R$. But this is not what is done in practice for masked pre-training, where normally all tokens in $R$ are used. It would have been interesting to see some discussion/experiments about the effect.*
>
> We understand that from a first sight, the main result provided in the l.h.s. of **Proposition 1** seems to indicate that we are conditioning on different subsets of remaining tokens $\boldsymbol{x}_{\mathcal{R}}$. As the reviewer indicates, this would not fit with MPT, that in practice uses all remaining tokens $\mathcal{R}$.
>
> However, the last expression in **Proposition 1** indicates a slight different thing. Particularly, the variable $x_{\mathcal{R}(1:D-j)}$ does not indicate a selection of subsets but the number of the remaining tokens instead. This is due to the sum in this proof considers different ratios between the masking size $M$ and the rest of tokens $R$. (See the sum which goes from $j=1$ until $M$). In the end, this corresponds to averaging over the number of the masked tokens $M$, where $R$ is always set up to the rest of items accordingly.
>
> Perhaps this point can be easily observed in the expression in the r.h.s. of the previous equation where we condition on all the remaining tokens in $\boldsymbol{x}_{\mathcal{R}}$. This one is also the one that matches Equation (1), building the final link with MPT. We apologise for this point that might lead to a little confusion. We would be happy to update this detail, also keeping the notation uncluttered, if the paper moves forward.
>
> **Point 2**
> > *It would have been interesting to see some experiments about the ideas stated in line 295.*
>
> Thanks for pointing this out. We have now included new experiments that perform what is described in L295. In particular, we  uniformly sample the number of masked tokens to obtain an unbiased estimate of the LML with the MPT losses. This generates what the theory indicates, the MPT losses are now centered with respect to the LML and they also make it converge to the true LML. Additionally, we also provide two extra Figures where we sample uniformly between the 0--50% and 50--100%. Obviously, as it is not sampling the entire range of the mask size, provides again biased estimates as we could have guessed.
>
> An important detail is that we initially discarded the uniform sampling due to it might become prohibitive on the computational cost. This could happen if the drawn masking rate is, for instance, around the 0%-5%, due to the model needs to recursively use all tokens at each iteration. Of course, this depends on the type of model chosen. But this is just for clarifying all details for the reviewer.
>
> The figures can be accessed in the additional PDF page attached to the general rebuttal response (to all reviewers).
>
> **Point 3**
> > *Given the works like Fong and Holmes (2020), there is no so much novelty from a theoretical point of view.*
>
> Perhaps, we have not correctly highlighted some the main theoretical differences with respect to the work of Fong and Holmes (2020) in our paper. Thus, to address the concern of the reviewer, we remark some important details that might help to understand better why the theoretical ideas brought in Fong and Holmes (2020) are significantly different from the ones shown in our submission.
>
> The work of *Fong and Holmes (2020)* introduced a first connection between leave-p-out *cross-validation* (CV) and the LML in such cases where CV uses posterior predictive probabilities as the scoring function. The equivalence is theoretically analysed from a Bayesian statistical perspective, where the main focus is placed on exhaustive leave-p-out CV and the prohibitive computational cost.
>
> Honestly, *Fong and Holmes (2020)* is a relevant reference because it inspired the probabilistic ML community to take a further look for understanding how LML is linked to many probabilistic settings. However, they successfully did it only for cross validation. In this way, we have found, proved and provided empirical results on the equivalence between MPT and LML for different models. This achievement was initially inspired on recent work like Chen et al. ICLR 2023, where Gaussian processes (GPs) are introduced in the architecture of transformers.
>
> ```
> W. Chen and Y. Li, Calibrating Transformers via Sparse Gaussian Processes. ICLR 2023
> ```
>
> **Point 4**
> > *Why does the LML change with the number of epochs?*
>
> If we guess correctly, the reviewer makes reference to the maximisation of the LML in the experiments and particularly, on the curves drawn in Figure 2. This plot shows three curves: the LML of the true model which generated the data, the cumulative MPT loss under analysis and the LML. The latter corresponds to the log-probability in Equation (6) for the PPCA model. In this experiment, we also maximise this expression of the LML to understand how it behaves compared with the MPT loss. Thus, the LML changes with the number of epochs as we are optimising wrt. the parameters $\boldsymbol{W}$ in L138 and L141. Interestingly, notice that both the LML and the MPT losses converge to the true value of the LML in the left hand side plot.
>
> **Point 5**
> > *What is GT-LML?*
>
> Good catch. GT-LML makes reference to *ground truth LML* in the caption of Figure 2, but we used the sentence "true model LML" in the label of the plots for the dashed line. It is a typo that will be corrected in the main paper.

---

> ### Comment · Reviewer_mVT4 · 2023-08-14
>
> Thank you very much for the clarifications. I have three final comments:
> - I do not think that Proposition (1) is a main contribution, since it can be directly derived from Proposition (2) in Fong and Holmes (2020).
>
> - The same discussion as in "3.2. Sensitivity to the prior and preparatory training" of Fong and Holmes (2020) also applies here:
> It might actually not(!) be beneficial to try so sample over all possible mask sizes, since
> this would include for example the case where R is empty or contains only one token, which means we basically ignore any context information.
>
> - By the way, I think Equation (1) is wrong:
> Equation (1) assumes that the masked tokens are independent conditional on rest R, but this assumption is obviously not valid.
> (In contrast, in Fong and Holmes (2020), when replacing x with iid samples y, this is valid.)

---

> > ### Author Response · Authors · 2023-08-14
> > **Response to additional comments from Reviewer mVT4**
> >
> > We thank the reviewer for the response to our clarifications. We would be happy to remark some important points around the final concerns indicated.
> >
> > >*I do not think that Proposition (1) is a main contribution, since it can be directly derived from Proposition (2) in Fong and Holmes (2020).*
> >
> > In the paragraphs L27-32 and L33-37 of our work, we clearly state how both propositions and proofs are connected. Initially, we mention that our derivations rely on a previous result from Fong and Holmes (2020), and we even changed the notation from $R$ to $M$ to make the reader understand better that cross-validation is in this case close to masked pre-training. The text in the paper literally says:
> >
> > *"Our proof relies on a previous observation from Fong and Holmes (2020), who shows that the log-marginal likelihood equals the average of exhaustive leave-M-out cross-validation (CV) given posterior predictive scores."*
> >
> > After that sentence in the same paragraph, we additionally explain that our formal results can be seen as the transposed version of Fong and Holmes (2020) results: where cross validation removes *random observations*, masked pre-training removes *random features*. In that way, we would like the reviewer to re-consider the statement *"it can be directly derived from Proposition (2) in Fong and Holmes (2020)"*. How is then possible to derive one result from the other? Particularly if Fong and Holmes (2020) is considering cross-validation and prediction of observations, while we are on masked pre-training and predicting random features of observations.
> >
> > In this way, we think that we have been completely transparent and trustworthy on the main contributions. Moreover, we have done our best to explain to the reader what the proof relies on and the main differences.
> >
> > > *The same discussion as in "3.2. Sensitivity to the prior and preparatory training" of Fong and Holmes (2020) also applies here (...)*
> >
> > Again, we understand the concerns around the connection with *Fong and Holmes (2020)*. We want to emphasize that we are big fans of the work of *Fong and Holmes (2020)*, but note that the link to the present paper is in the form of a *proof technique*. They do not touch upon MPT at any point, so a similar reasoning on the experiments and analysis does not imply it to be "the same".
> >
> > Moreover, we think as in the previous point, that this sort of statements are not well founded given what is presented in the paper. The interest of *Fong and Holmes (2020)* is very different there, as they discuss the prior and the number of data-points (objects not features) to be considered in cross-validation and not the amount of masked tokens as we do. In our case, we  wanted to understand what was going when one fixes the masking rate, which is a very different perspective.
> >
> > > *By the way, I think Equation (1) is wrong (...) In contrast, in Fong and Holmes (2020), when replacing x with iid samples y, this is valid.)*
> >
> > We remark that Equation (1) belongs to section **2 Masked pre-training** where we make a short background for the reader. Here, nothing around *Fong and Holmes (2020)* applies yet. Thus, Equation (1) is not wrong, as we clearly explain what is being maximised by MPT. In this case, if the masked tokens $x_{\mathcal{M}}$ are conditionally independent given the rest $x_{\mathcal{R}}$, it makes sense that the conditional distribution factorises according to a sum. It is important to understand here that the Equation is showing how the variables are modelled when building the objective. If the reviewer considers that placing there a $\approx$ symbol in the equality, we would be happy to update it. One last detail for reference is that this sort of notation was previously used in the context of MPT. Particularly in Song et al. NeurIPS (2020) @ Equation (1) and Yang et al. NeurIPS (2019) @ Eq. (3).
> >
> > ```
> > Z. Yang , Z. Dai, Y. Yang , J. Carbonell, R. Salakhutdinov, Q. V. Le, XLNet: Generalized Autoregressive Pretraining for Language Understanding, NeurIPS 2019
> > ```
> >
> > ```
> > K. Song , X. Tan , T. Qin , J. Lu and T. Y. Liu, MPNet: Masked and Permuted Pre-training for Language Understanding, NeurIPS 2020
> > ```

---

### Official Review · Reviewer_EYbT · 2023-07-10

**Soundness:** 4 excellent
**Presentation:** 4 excellent
**Contribution:** 4 excellent
**Rating:** 6
**Confidence:** 4

**Summary:**

This paper proposes viewing masked pretraining methods (like BERT or MAE) as optimizing a biased form of the log marginal likelihood $\log p_\theta(x)$. Using a parallel to exhaustive leave-M-out cross validation, the authors derive the exact relationship between the LML and a weighted sum of masked pretraining losses across mask sizes. The authors show that maximizing the masked pretraining loss improves the LML, across a wide variety of settings, from probabilistic PCA to VAEs to BERT.

**Strengths:**

- I found Prop. 1, the connection between LML and the weighted sum of masked pretraining losses, to be original and quite enlightening.
- Paper is thorough and provides extensive empirical experiments to confirm the theory.
- Finding seems like a good step towards understanding masked pretraining objectives, which are extremely widespread in both NLP (BERT) and vision (MAE), and potentially understanding how to improve them.
- Paper is well written and clear, especially if the appendix is used to supplement the derivations in Section 3.

**Weaknesses:**

- The biggest issue is that the paper shows that masked pretraining performance improves LML, but does not address why it does better than other training objectives that directly maximize LML (such as an autoregressive model).
- No experiments on the vision side, even though MAE is an extremely popular method that falls neatly into the framework.
- Some parts of the paper are a tiny bit confusing. Surprisingly, I felt that Appendix A and B could be swapped into the main paper and it would significantly improve the clarity of Proposition 1's derivation or the PPCA setting.
  - I felt that the second paragraph of the intro (L19-26) didn't sufficiently motivate why the LML is important.

**Questions:**

**Major questions**:
- If maximizing LML is the ultimate reason why MPT is good, then why are other methods of directly maximizing LML (e.g., autoregressive models) worse?
- Can you run the same BERT experiments with pretrained MAE models as well, to show applicability to vision models?

**Minor questions and suggestions**:
- Where did this "log-marginal likelihood" name come from? It looks just like the $\log p_\theta(x)$ that all generative models maximize, so I'm confused about the naming.
- L19-26: This paragraph doesn't really convince me that the marginal likelihood is that important.
- L27-28: "masked pre-training optimizes according to a stochastic gradient of the log-marginal likelihood" -- I think this sentence should be removed, since the following sentence states the contribution more clearly.
- L33: "who shows" -> "who show"
- L74: "which might also include likelihood" -- what is likelihood?
- L86: "previous sum" -- is this referring to Eq. 2?
- L87-92: I understand what it's trying to say after reading the entire paper, but this paragraph is very confusing.  What do exchangeable and invariant mean?
- The block of text between L108-109: "to be aligned" -- this is also confusing. It's explained much better in the appendix!
  - Generally, Appendix A explains this proof much better than the main text.
- L128: "we usually have a biased estimator" -- wouldn't a Monte Carlo estimator, with masks drawn from the right distribution, be unbiased?
- L157: "just one random mask per training epoch" -- my understanding is that masks are different for each sample in each epoch. I'm reading this as using the same random mask for all samples in a given epoch.
  - L165: "notice that P=1 is usually set up in MPT in practice" -- same issue as above. What exactly is being claimed about the number of masks?
- L170: "this bias is known" -- known in what sense? The numerical value of the bias, the symbolic form of the bias, or just the fact that the MPT objective is not equal to the LML objective?
- L183: "the exact LML is iteratively maximized at each epoch" -- I wouldn't use the word "maximize" here, as it seems like it increases but is not guaranteed to even reach a local minimum unless there are further assumptions on the structure of the model.
  - Same issue with L204-205
- Fig. 2 caption: "The rate of masking is unfixed and it varies from 1% to 100%" -- I may be misunderstanding here, but with 10 tokens, shouldn't the masking rate start at 10%?
- L194-195: probabilities imply integrating out the latent variables, often given the posterior distribution. In most cases, performing this integration is extremely difficult or not possible in training time." -- why do there need to be latent variables at all? Latent variables are one way to induce a structured joint distribution over all dimensions of x, but this doesn't have to be used. In fact, latent variables aren't used in the proof of Prop 1.
- L202: "as well as we compare" -> "as well as compare"
- L209-210: "we used iterative encoding-decoding to produce the self-conditional probabilities for masked tokens — see Sec. F in Rezende et al. (2014)" -- it would be good to explain this procedure here. I glanced at Sec F and it seems like it only gives a way to sample from the marginal over masked tokens, not a way to compute the probability? I may be misunderstanding, which is why it would be nice to have a clear explanation in this paragraph.
- L212-214 on implicit integration: these lines are confusing and would benefit from being rewritten.
- Fig. 3 caption: "Darker lines indicate the evolution of the LML" -- the figure legend does not match this. Also, i don't understand why the lower right plot is separate from the lower left plot, or why the value is so different.
- Fig. 4: "looses" -> "loses." Also, I don't understand what the black lines in Fig. 4 (right) is supposed to denote.
- L218-219: "This explains why the probabilities have an approximately exponential decay" -- I don't see how anything explains this.
- L224: "effect to" -> "affect"
- L225: "a longer discussion is provided in the supplementary material" -- Appendix C.2 doesn't seem very relevant to this section.
- Table 2 caption: "area under the MPT curve" -- is this weighted by the number of masks or not? Wouldn't the weighted sum based on Prop. 1 directly measure LML here?
- Fig. 5: y axis label is cut off on the far left plot.





**Limitations:**

A major limitation that isn't addressed by the authors is this: why is MPT the right way to improve LML, instead of another method that directly maximizes LML? Empirically, masked pretraining methods like BERT/MAE are much better representation learners than alternate approaches that optimize the LML. The authors mention "autoregressive modeling" in their limitations section, but I don't think enough weight is put on it as a large question that this paper on MPT<-> LML introduces yet doesn't answer.

---

> ### Author Rebuttal · Authors · 2023-08-09
>
> We thank the reviewer for the positive consideration and the useful feedback on our work. We are particularly glad for the scores marked as *excellent* and for hearing that the connection between LML and the weighted sum of masked pretraining losses is considered *original* and *enlightening*. Overall, we appreciate the time you clearly took to review our paper, including the key questions and the rest of minor details attached. We've done our best to fit in this rebuttal clear answers to the principal points of your review. Additionally, we have performed extra experiments on the vision side with MAE to address one of the reviewer's main concerns.
>
> **Point 1A** & **1B**
> > *If maximizing LML is the ultimate reason why MPT is good, then why are other methods of directly maximizing LML (e.g., autoregressive models) worse?*
>
> > *why is MPT the right way to improve LML, instead of another method that directly maximizes LML? (..)*
>
> Our ambition is not to say that maximising the LML is the **ultimate** reason why MPT works so well. Perhaps we need to revise our communication if a final update of the paper is allowed.
>
> Instead, we are strictly providing a positive answer to the question: *Is MPT related to the maximisation of LML?* The complete answer included in the paper is that MPT optimises according to a stochastic gradient of the LML. As the latter is a well-known measure of generalization ability in Bayesian models, we believe that the connection provides a new tool for analysing MPT. However, we do not claim it to be the unique reason, as MPT is a learning algorithm that might be combined with different models.
>
> In the particular case of LLMs, we should not ignore the role that the chosen architecture for the model plays. One additional example that supports this point is the work of (Neyshabur, ICLR 2019), which says the following: *Our capacity bound correlates with the behavior of test error with increasing network sizes (...), and could partly explain the improvement in generalization with over-parametrization*. Even in this example, the explanation is assumed to be *partial*, and our perspective is similar in this way.
>
> With respect to the *direct maximisation of LML*, we cannot claim that it is generally *worse* than MPT. However, we have observed that MPT maximises according to a stochastic approximation to the training objective. From a practical perspective, this gives advantages with respect to the direct maximisation of the objective, where the lower computational cost excels and the stochastic optimization might be beneficial to avoid local minima.
>
> **Point 2**
> > *Can you run the same BERT experiments with pretrained MAE models as well, (..)*
>
> Yes. In particular, we reproduced the same empirical results obtained in **Section 3.2** with BERT, but for the MAE approach instead. We considered the exact model included in He et al., CVPR 2022. The results are shown in the additional PDF attached to the global response.
>
> ```
> K. He, X. Chen, S. Xie, Y. Li, P. Dollar and R. Girshick, Masked Autoencoders Are Scalable Vision Learners, CVPR 2022
> ```
>
> **Point 3**
> > *The authors mention "autoregressive modeling" (..) but I don't think enough weight is put on it (..).*
>
> Fair point. We understand the interest around *autoregressive modelling* and its connection with MPT. However, we do think that this topic slightly falls out of the scope of our work, as our main goal was to build the link between LML and MPT in a thorough manner.
>
> Importantly, autoregressive modelling is mentioned in the discussion section to point out that there is still room for understanding the link with LML beyond the simplest version of MPT. This is not really a limitation, but rather potential future work. Our early hypothesis is that the theory proposed could accept conditioning between the tokens to-be-predicted. This could make the pre-training algorithm less of an approximation to the true LML. However, further analysis and theoretical development should be needed to prove this crucial point.
>
> **Point 4.1**
> > *Where did this "log-marginal likelihood" name come from? (...).*
>
> The *marginal likelihood* name has been longly used in the Bayesian ML for more than 25 years, at least. In his seminal works around 1998 with Bayesian NNs and Laplace approximations, David JC MacKay already mentions marginal likelihood as a method for model comparison also providing even earlier references. A common synonym for marginal likelihood is *evidence*, which also gives name to the acronym ELBO (Evidence Lower BOund) widely used in variational inference and VAEs, for instance. This probability is indeed the one being bounded by the ELBO.
>
> ```
> D. JC MacKay. Choice of Basis for Laplace Approximations. Machine Learning, volume 33, pp. 77–86, Springer, 1998
> ```
>
> **Additional comment:** We would've liked to include the rest of answers to the questions in the minor comments and suggestions (those marked as point 4.1, 4.2, etc..). Unfortunately, we did not have enough space in this rebuttal box. As soon as the discussion phase will begin, we will include the mentioned answers in an additional comment for the reviewer.

---

> > ### Author Response · Authors · 2023-08-11
> > **Response to additional questions (Part 1/2)**
> >
> > As we mentioned in the main rebuttal, we include the answers to the additional questions included in the section of minor points of the Reviewer **EYbT**. We hope that it helps to address all remaining concerns and we thank again for the time taken to review our work.
> >
> > **Point 6.2**
> > > *L87-92: (...). What do exchangeable and invariant mean?*
> >
> > Here the words **not exchangeable** make reference to the fact that we cannot exchange different conditional probabilities freely, for instance, $p(x_1|x_2,x_3)\neq p(x_2|x_1,x_3)$. This might seem trivial for the reader, but certain statistical models use *exchangeability* as a key property between observations. Thus, we wanted to highlight that this is not the case in our work. On the other hand, **invariant** makes reference to the sum of conditional factors, whose result is always the same and equal to the value of the LML.
> >
> > **Point 6.3**
> > > *L128: "we usually have a biased estimator" -- wouldn't a Monte Carlo estimator, with masks drawn from the right distribution, be unbiased?*
> >
> > Yes. To avoid having an unbiased estimator of the LML, using Monte Carlo sampling would be a good idea. However, as we know that the size of masking is usually fixed in practice, we focused on understanding what was going on in that case.
> >
> > As this question was also raised by the reviewer **mVT4**, we performed additional experiments to empirically check that this is indeed true. One important detail to mention is that we initially discarded this strategy due to it might become prohibitive on the computational cost. If we consider uniform sampling of mask sizes, and the resulting rates are around 0%-5%, the computational cost of considering all tokens might be unfeasible in some types of models.
> >
> > The experimental results are also included in the additional PDF attached to the global response to all reviewers.
> >
> > **Point 6.4**
> > > *L157: "just one random mask per training epoch" -- my understanding is that masks are different for each sample in each epoch. (...).*
> >
> > > *L165: "notice that P=1 is (...) What exactly is being claimed about the number of masks?*
> >
> > Good catch. Perhaps this sentence is not clear enough without the proper context. What we would like to say with *just one random mask per training epoch* comes from the following:
> >
> > **Proposition 1** tells us that we are equivalent to the LML, if we average over **all** the different masking patterns. In practice, this mean that we would have to sum over the total number of choices $\mathcal{C}_M$, which is unfeasible as the number prohibitively increases with the number of tokens per observation. In that sense, we assumed that the expectation in Proposition 1 is approximated with less masking patterns (e.g. one mask). We analysed this effect in the experiments shown in Figure 1, where we vary with the *number of random masks* ($x$-axis).
> >
> > The main consequence of this analysis is that averaging in **Proposition 1** with only *one* random mask is a useful approximation in this scenario. That's why we mention that.
> >
> > **Point 6.5**
> > > *L170: "this bias is known" -- known in what sense? (...)*
> >
> > The *bias* of the estimate is given by the difference between the cumulative log-probabilities for all sizes of the mask and the log-probabilities obtained only with one fixed mask size. Additionally, this bias is constant, which is an important property for training the model. This bias can be observed in the distance between the shaded curves and the dark line ones in Figure 2. Additionally, it could be computed if needed.
> >
> > **Point 6.6**
> > > *Fig. 2 caption: "The rate of masking is unfixed and it varies from 1% to 100%" -- I may be misunderstanding here, but with 10 tokens, shouldn't the masking rate start at 10%?*
> >
> > Good catch. It looks like a typo. We will correct it in the updated version.

---

> > ### Author Response · Authors · 2023-08-11
> > **Response to additional questions (Part 2/2)**
> >
> > **Point 6.7**
> > > *L194-195: probabilities imply integrating out the latent variables, (...)." -- why do there need to be latent variables at all? Latent variables are one way to induce a structured joint distribution over all dimensions of x, but this doesn't have to be used. In fact, latent variables aren't used in the proof of Prop 1.*
> >
> > Fair point. We used latent variables as a way to link the theoretical results with the tractable PPCA model. In this case, posterior predictive probabilities are obtained via integration of such latent space. We were also interested in connecting our analysis with previous works as Lucas et al. NeurIPS 2019. We agree that the proof holds without considering latent variables explicitly as it uses directly log-posterior predictive probabilities. In some way, it has been our choice for the probabilistic model in this work, but others would also be possible.
> >
> > ```
> > J. Lucas, G. Tucker, R. Grosse and M. Norouzi. *Don’t Blame the ELBO! A Linear VAE Perspective on Posterior Collapse*. NeurIPS 2019
> > ```
> >
> > **Point 6.8**
> > > *L209-210: "we used iterative encoding-decoding to produce the self-conditional probabilities for masked tokens — see Sec. F in Rezende et al. (2014)" -- it would be good to explain this procedure here.(...)
> >
> > Fair point. Obtaining conditional distributions for VAEs is not a trivial task. Some works in the recent literature have addresses this problem, but for MPT we were interested in the simplest way to do it at every training step.
> >
> > In his seminal work, Rezende et al. 2014 (Appendix F) propose a way to do missing imputation (e.g. non-observed pixels) using the VAE model. We reproduced the following steps described in their appendix:
> >
> > *"(i) initializing the non-observed pixels with random values; (ii) sampling from the recognition distribution given the resulting image; (iii) reconstruct the image given the sample from the recognition model; (iv) iterate the procedure."*
> >
> > We did the same in our experiments with our own data. This process converges to the desired conditional distribution (named conditional as it is conditioned on the observed entries $\mathbf{v}_o$). In the Appendix it is clearly stated, and named as marginal, but corresponds to our predictive distributions used for MPT. They indeed give the following sentence:
> >
> > *"conclude that given the above properties, (...) is guaranteed to generate samples from the correct marginal $p(\mathbf{v}_m|\mathbf{v}_o)$."*
> >
> > Moreover, thanks for point this out, we will add extra details for improving the clarity on the reproducibility of these experiments.
> >
> > ```
> > D. J. Rezende, S. Mohamed and D. Wierstra. Stochastic Backpropagation and Approximate Inference in Deep Generative Models. ICML 2014
> > ```
> >
> > **Point 6.9**
> > > *Fig. 3 caption: "Darker lines indicate the evolution of the LML" -- the figure legend does not match this. (...).
> >
> > The reviewer is right, this should be corrected. Thanks for pointing it out, we will update the plots if the paper moves forward.
> >
> > **Point 6.10**
> > > *Also, I don't understand what the black lines in Fig. 4 (right) is supposed to denote.*
> >
> > Right. We did not use the necessary amount of text here to explain these lines. Simply, we wanted to highlight the impact of choosing 15% or 85% rate in the masking procedure. When doing this, we are indeed approximating the area under the curve with a constant mass (the areas under the line). Interestingly, in some cases this area is larger or smaller, depending on which is the rate of masking chosen.
> >
> > **Point 6.11**
> > > *"This explains why the probabilities have an approximately exponential decay" -- I don't see how anything explains this.*
> >
> > In the analysis of the results shown in Figure 4, we were making reference to the (approximate) exponential decay of the curves as we increase the rate of masking to >90%. In practice, this means that the model does not have enough number of tokens to accurately predict the masked ones. That's why curves describe the low predictive probabilities.

---

### Author Rebuttal · Authors · 2023-08-09

**General comment to all reviewers**

We thank all reviewers for their useful comments, positive consideration and relevant feedback on our paper. It seems that the reviews are positive in general and acknowledges our main contributions and soundness of our work. We have addressed each comment and question individually below and we would be glad to engage in discussion in case of more questions or concerns exist. We also provided an extra PDF with the new results from the experiments that some of the reviewers asked for. We will also update the main paper in the future with the main requested changes for improvement, if the submissions moves forward.

Moreover, as some reviewers already did in their comments, we use the acronyms MPT (masked pre-training) and LML (log-marginal likelihood) in our response.

**Note on the additional experiments**

Additionally to the main response in this platform, we provide the results for the suggested experiments in the attached PDF. In particular, the plots included in Fig. 1 and the response **E.1** addresses the **Point 2** of the reviewer **mVT4**. Additionally, the results shown in Fig. 2 and the section **E.2** correspond to the **Point 2** of the reviewer **EYbT**.

**E.1.  Experiments on uniform masking rate sampling**

These experiments answer the question around the effect of uniformly sampling the masking ratio with MPT losses. So far, we have observed that the cumulative MPT loss is equivalent to an \*unbiased* estimate of the log-marginal likelihood (LML) when we consider all possible numbers for the amount of masked tokens. To avoid having a *biased* estimate when fixing the masking rate (e.g. to 20\%), one option is to use an uniform distribution. In this way, we sampled the rate of masking at each epoch in the range ($0\%-100\%$). The results in the (A) plot of Fig. 1 indicate that we are able to obtain such unbiased target loss. Importantly, we should notice that the cumulative losses oscillate around the true value of the LML that is also maximised. For completeness of the experiments, we also included the empirical results when the masking rate is sampled in different ranges (i.e. $50\%-100\%$). In this case, we have different biases in the plots (B) and (C) in the Fig. 1. This biases are related to the area under the curves described in Fig. 4 of the main paper.

**E.2. Experiments on vision.**

Additionally to the results included in Fig. 5 of the main paper with the BERT model, we now provide the curves produced in our experiments with the ViT-MAE model. For this study with vision self-supervised learners, we have loaded a pre-trained model of ViT-MAE from a public repository. To draw the curves in Fig. 2, we computed the losses per each rate of masking between 5% and 95% for samples from three different test image datasets. We can observe that the curves described by the pre-trained ViT-MAE model shows a similar behaviour to the one we obtained with  BERT. These curves are also aligned with the *canonical* analysis performed with the tractable models. On the hand, the curves described by ViT-MAE with randomly initialised parameters are *flat* and of low self-predictive probability.

---

### Comment · Area_Chair_RdaK · 2023-08-21
**Mixed reviews**

Dear reviewers,

Thanks for the hard work so far!

This paper received mixed reviews and the authors have responded multiple times.

We need to ideally reach a consensus in the rebuttal period, and at least should have active discussions, updated recommendations, and acknowledgment that you have read the response.

Can you please check the reviews (especially the ones who disagree with you) and the author rebuttal and see if your opinion has changed? Please give your reasoning in as much detail as possible.

AC

---

### Decision · Program_Chairs · 2023-09-21

**Decision:**

Accept (poster)

**Comment:**

The paper received mixed reviews leaning toward acceptance (1 accept, 2 weak accepts, and 2 borderline rejects) after the rebuttal period. The AC checked the reviews and discussions, and believes the reviews have covered most aspects of the paper -- some checked the proofs in detail, others relied more on intuitive understandings. Overall, while the paper has some claims and descriptions that are not worded ideally, it establishes an interesting connection between masked pre-training and the maximization of log-marginal likelihood. This is by itself a valuable theoretical contribution, and the authors also did additional experiments during the rebuttal to enhance the empirical justifications of the work. Therefore recommending acceptance -- the authors are expected to incorporate useful feedbacks from the reviews for the final version.